# Instance Segmentation Method for Insulators in Complex Backgrounds Based on Improved SOLOv2

**DOI:** 10.3390/s25175318

**Published:** 2025-08-27

**Authors:** Ze Chen, Yangpeng Ji, Xiaodong Du, Shaokang Zhao, Zhenfei Huo, Xia Fang

**Affiliations:** 1State Grid Hebei Electric Power Research Institute, Shijiazhuang 050000, China; scu2023dujiaqi@163.com (Z.C.);; 2State Grid Hebei Electric Power Co., Ltd., Shijiazhuang 050031, China; 3School of Mechanical Engineering, Sichuan University, Chengdu 610065, China

**Keywords:** SOLOv2, HRNet, transmission inspection, instance segmentation, NSCT

## Abstract

To precisely delineate the contours of insulators in complex transmission line images obtained from Unmanned Aerial Vehicle (UAV) inspections and thereby facilitate subsequent defect analysis, this study proposes an instance segmentation framework predicated upon an enhanced SOLOv2 model. The proposed framework integrates a preprocessed edge channel, generated through the Non-Subsampled Contourlet Transform (NSCT), which augments the model’s capability to accurately capture the edges of insulators. Moreover, the input image resolution to the network is heightened to 1200 × 1600, permitting more detailed extraction of edges. Rather than the original ResNet + FPN architecture, the improved HRNet is utilized as the backbone to effectively harness multi-scale feature information, thereby enhancing the model’s overall efficacy. In response to the increased input size, there is a reduction in the network’s channel count, concurrent with an increase in the number of layers, ensuring an adequate receptive field without substantially escalating network parameters. Additionally, a Convolutional Block Attention Module (CBAM) is incorporated to refine mask quality and augment object detection precision. Furthermore, to bolster the model’s robustness and minimize annotation demands, a virtual dataset is crafted utilizing the fourth-generation Unreal Engine (UE4). Empirical results reveal that the proposed framework exhibits superior performance, with *AP*0.50 (90.21%), *AP*0.75 (83.34%), and *AP*[0.50:0.95] (67.26%) on a test set consisting of images supplied by the power grid. This framework surpasses existing methodologies and contributes significantly to the advancement of intelligent transmission line inspection.

## 1. Introduction

Transmission lines are persistently subjected to the elements of the natural environment, making them susceptible to a myriad of potential risks, such as those posed by avian interference, ice accumulation, arboreal obstructions, external physical damage, strong wind forces, and landslides. Each of these factors may profoundly affect the safe operation of these lines. Consequently, the protection of transmission lines is of paramount importance. The principal aim of transmission inspection is to ensure the uninterrupted functioning of power equipment, maintain electrical safety, and preserve the stability and reliability of power generation and supply processes. Through meticulous inspection, the incidence of power outages can be significantly diminished, latent hazards can be promptly detected, and the likelihood of severe accidents or the escalation of equipment maintenance costs due to malfunctions can be effectively mitigated.

Insulator damage is a prevalent issue in transmission line inspection [1], as it can give rise to hidden dangers such as widespread faults in the transmission lines. However, detecting insulator damage presents significant challenges due to its small size and the fact that it is considered a minor defect. This poses a great challenge for object detection algorithms, as the proportion of the target (insulator damage) within the entire image is extremely small.

Furthermore, the background environment in inspection images is often complex and diverse, and the angles at which drones capture these images can vary. Additionally, factors such as mutual occlusion between other transmission components further complicate the detection of insulator damage defects.

The defect detection methods for insulator breakage can be divided into two main categories. The first category is based on traditional image processing anomaly detection algorithms: Lv et al. [2] proposed an image-based method to detect insulators using image tracking technology and selecting the segmentation threshold based on the histogram envelope. Reddy et al. [3] proposed the use of Discrete Orthogonal S-transform (DOST) combined with intelligent classification algorithms to detect the state of insulators in 11 Kv distribution lines. Zhai et al. [4] built a model based on the color characteristics of insulators and combined it with spatial features to identify the target region of insulators. Potnuru et al. [5] proposed obtaining statistical features of each insulator using statistical features of the curvilinear wave transform and the local ternary pattern (LTP) histogram.

These methods aim to detect insulator breakage using traditional image processing techniques. However, they may have limitations in terms of recognition accuracy, robustness in complex environments, and processing time.

The second category of defect detection methods is based on deep learning techniques, which have shown promise in addressing the limitations of traditional image processing methods. These methods have demonstrated good recognition effects in experimental environments and offer potential solutions to the challenges faced by traditional approaches, including low recognition accuracy, poor robustness in complex and changing environments, and long processing times.

Gao et al. [6] proposed a deep learning-based method for insulator explosion recognition to detect faulty insulators. Cui et al. [7] introduced a semantic segmentation algorithm based on multi-scale feature fusion to achieve insulator segmentation in different environments. Wei et al. [8] proposed a multi-model fusion computational method for transmission line insulator self-explosion detection based on edge computing and deep learning. Li et al. [9] utilized a region-based fully convolutional network (R-FCN) deep learning algorithm to realize defect detection of insulators.

Although these studies have produced certain outcomes in identifying defects in high-voltage transmission line components, notable constraints persist. Images captured by UAVs frequently display intricate backgrounds, large dimensions, and variable insulator scales, necessitating considerable effort for dataset annotation.

The method is proposed with an instance segmentation framework based on improved SOLOv2 [10]. HRNet [11] was used as the backbone to better extract multi-scale image information while retaining the overall framework of SOLOv2. In order to improve the quality of the mask, a Non-Subsampled Contourlet Transform (NSCT)-based image contour preprocessing channel was integrated, and CBAM was introduced. In addition, we increased the input size to 1200 × 1600. Finally, in order to enrich the dataset, improve the generalization performance of the model in complex backgrounds, and reduce annotation work, a virtual dataset created by UE4 was trained together with the real dataset.

The innovative contributions of this article are as follows:The insulator segmentation approach, enhanced by an improved SOLOv2 algorithm and applied to large-scale UAV imagery, demonstrates effectiveness in accurately segmenting insulators, thereby deriving their masks and contours.Incorporating an additional input channel dedicated to NSCT image contours enriches the input data, consequently enhancing the quality of the resultant masks when used in conjunction with CMBA.The virtual dataset generated through UE4 effectively diminishes the workload associated with annotation tasks, augments the dataset, and enhances the model’s generalization capabilities.

## 2. Proposed Framework

### 2.1. Improved SOLOv2

SOLOv2, a deep learning model utilized for object detection and instance segmentation, has made significant improvements over its predecessor SOLO [12] (Segmenting Objects by Locations). It has demonstrated strong performance in numerous instance recognition tasks [13,14,15,16].

The improved SOLOv2 builds upon the fundamental concept of SOLO, which states that target instance segmentation can be achieved by directly predicting the category of each pixel and accurately localizing it within a specific target instance. Its structure is shown in Figure 1. In SOLOv2, a Fully Convolutional Network (FCN) and a Feature Pyramid Network (FPN) are used to extract the feature of the image and obtain feature maps at multiple scales (SOLOv2’s original backbone is Res50 [17]).

These feature maps have a shape of ((H) × W × E), where H, W, and E represent the height, width, and number of channels, respectively. Notably, the H and the W are 1/4 of the input image’s height and width, respectively.

The subsequent detection header consists of two branches: the category branch and the feature branch.

In the category branch, the image is divided into S × S regions. When the center of a predicted category object falls within a specific region, its position needs to be predicted. As a result, the output dimensions are S × S × C, where C represents the total number of categories that require prediction in the dataset.

Regarding the mask branch, the improved SOLOv2 adopts a dynamic convolution strategy. This branch is further divided into two sub-branches: the convolution kernel branch and the feature branch. In the convolution kernel branch, a series of four convolutional layers and one convolution layer with a 3 × 3 × D kernel are employed. This transforms the input into a 3 × 3 × D tensor. Following this, a feature map of size H × W × S is produced through convolution, resulting in the prediction mask G of size E × S × D. In the feature branch, the feature map of size H × W × E is transformed into a feature map of size H × W × E after convolution with a 1 × 1 kernel, group normalization, and Rectified Linear Unit (ReLU) operations. After adjusting the number of channels by 1 × 1 convolution, the corresponding region mask of feature F is obtained. Finally, the redundant prediction results are removed by Matrix Non-Maximum Suppression, and the segmentation regions of each instance are finally obtained.

The loss function of SOLOv2 is defined as follows:(1)$$L=L_{C}+{\lambda L}_{M}$$
where $L_{C}$ denotes the focal loss of traditional semantic category segmentation, which solves the problem of positive and negative sample imbalance that occurs during the training process, and $L_{M}$ denotes the loss function of mask prediction. Based on multiple experimental tests, when *λ* is 0.5, the convergence effect of the entire model is the best. To pay attention to small target work, we adopted the focal loss mechanism and processed the low-confidence mask in the manner of Formula (2).(2)$$L_{Focal}=(1-\alpha )L$$

Among them, α is the confidence level for classifying mask pixels. The higher the confidence level, the closer it is to 1, and the less information is passed back in the backpropagation.

### 2.2. Preprocessing

The Non-Subsampled Contourlet Transform (NSCT) is a mathematical transformation method utilized in image processing in Figure 2. It is an improvement upon the wavelet transform, with the goal of effectively capturing contour and texture information in images while providing capabilities for multi-scale, multi-directional, and polygonal representations. NSCT achieves this by decomposing images using wavelet filters at various scales and directions, thereby capturing texture information across different frequency bands and orientations. This work adopts the method of high-frequency enhancement. The 4/1 and 2/1 high frequencies have higher passing weights, while the low-frequency part reduces the transmission of data information and enhances the contour information.

The fundamental concept of NSCT involves decomposing an image into multiple frequency bands and extracting local features through contour wave transformation within each band. These local features encompass edges, textures, and other intricate details. By implementing image analysis and processing with NSCT, it becomes possible to extract a broader range of image features at different scales and directions, ultimately enhancing the accuracy and effectiveness of image processing. NSCT finds wide-ranging applications in the field of image processing, particularly in tasks such as edge detection, image enhancement [18,19], and image fusion [20]. Upon using NSCT to extract image contours and fusing them with the original image, the original three-channel image input is transformed into a four-channel representation.

### 2.3. HRNet as Backbone

HRNet is a high-resolution network architecture for computer vision tasks such as image classification, target detection, and image segmentation.

HRNet constructs a multi-branch parallel structure, each branch is responsible for processing feature maps with different resolutions, while consistently retaining a high-resolution branch (1/4 of the original input size), and information is exchanged and fused between these branches at each stage. The parallel structure allows HRNet to maintain high-resolution information while taking into account the multi-scale features’ expression ability. The improved HRNet network structure is shown in Figure 3. Since the input size of the image used in this paper is large, reaching 1200 × 1600, in order to ensure the sensory field of the network, a stage is added on the basis of the original structure of HRNet in order to obtain the information of a larger sensory field and increase the depth of the convolutional layer, and the number of modules in each stage is modified from [1,1,4,3] to [2,2,3,3,3] to control the scale of model parameters. In order to control the size of the model parameter number, and at the same time, due to the addition of a channel, which adds a large amount of useful information and reduces the dependence on the feature extraction capability of the network, the number of channels in each of its layers is reduced. Specifically, the number of channels in the feature maps corresponding to 1/4, 1/8, 1/16, and 1/32 of the original image was adjusted from 48, 96, 192, and 384 to 24, 48, 96, and 144, respectively. Additionally, the number of channels in the new feature map corresponding to 1/64 of the original image was set to 192.

### 2.4. Convolutional Attention Module CBAM

CBAM is an attention mechanism module designed for computer vision tasks. It can be seamlessly integrated into a Convolutional Neural Network (CNN) to significantly enhance the network’s ability to model spatial and channel attention. The CBAM module is comprised of two sub-modules: the Channel Attention Module, which assigns weights to features across different channels to extract the most representative features, and the Spatial Attention Module, which assigns weights to spatial locations in the feature map to highlight important regions. By leveraging this module, CNNs can efficiently capture both spatial and channel dependencies, leading to improved performance in computer vision tasks.

### 2.5. Select the Confidence Interval

Upon traversing the specified framework, enhancement edge data is derived within the Semantic Branch of SOLOv2, whereas pixel-level classification data is extracted from the Category Branch. In the Semantic Branch, edge pixel information is assigned a greater significance, playing a pivotal role in determining its affiliation with the overarching segmentation feature. Meanwhile, feature value attribution is determined based on weighted calculations within the Classification Branch. As illustrated in Figure 4, the mask presenting the highest confidence within the Semantic Branch undergoes selection via non-maximum suppression. It is subsequently amalgamated with classification pixels in the Category Branch to acquire the edge with the highest confidence alongside pixel segmentation that yields the highest classification accuracy.

## 3. Experimentation

### 3.1. Experimental Environment

The experimental platform for this article used the Ubuntu 22.04 operating system. The framework selected was the open-source project mmdetection from Openmmlab. The core environment configuration included Python 3.8, PyTorch 1.12.1, CUDA 11.3, mmdet 2.25.1, MMCV 2.1.0, and NumPy 1.21.6. To enhance model training, a NVIDIA GeForce GTX 2080 Ti GPU with 22 GB of graphics memory was utilized.

### 3.2. Data Sources

The data used for the experiments in this paper consists of two parts: the real dataset and the virtual dataset created by UE4.

The real dataset is the high-resolution images captured by the high dynamic range (HDR) camera on the UAV, which was manually operated by the grid inspection personnel in the high-voltage transmission inspection task. The resolution is 3648 × 4864 (aspect ratio 3:4) and 3648 × 5472 (aspect ratio 2:3). We selected 4099 suitable images for training, validation, and test sets, which were divided into a ratio of 7:1:2. This dataset contains three types of insulators—glass, porcelain, and composite—and contains 6241 insulator instances of various scales, and the samples are shown in Figure 5.

The virtual dataset used in this study was created using Unreal Engine 4 (UE4) and a cross-platform simulator called AirSim. Researchers were able to utilize the onboard camera of the virtual UAV to record images within the simulated environment. Furthermore, the corresponding label data for each frame could be directly exported, which significantly reduced the time required for dataset creation. To replicate a realistic scenario, a simulated transmission line was constructed within a broad mountain landscape simulation environment. This virtual environment closely resembled the layout of a real-life scene. The obtained data samples from this virtual dataset are depicted in Figure 6. The virtual dataset serves as a means to simulate real-world conditions to a certain extent. It can be utilized as a component of data augmentation to enhance the algorithm’s performance. By incorporating virtual data alongside real-world data, the algorithm can potentially improve its ability to generalize and perform well in varied environments.

### 3.3. Training Details

In the experiment, the optimizer used was Stochastic Gradient Descent (SGD). The initial learning rate was set to 0.001, and the weight decay coefficient was set to 0.0001. The momentum value was set to 0.9. The maximum number of iterations or epochs was limited to 100, and the batch size used was 2. As shown in Figure 7, since the images during the patrol flight are all collected by the unified patrol flight team, we adopt a processing method that combines the edge end with the cloud. On the processing card of the edge segment NT250, 1200 × 1600 pic 28 FPS can be achieved.

Regarding the input images, they were resized uniformly to 1200 × 1600 while maintaining the original image scale. Insufficiently covered areas were filled in with gray to ensure consistency. During the initial training stage, a linear warmup strategy was employed to gradually increase the learning rate. This helps to reduce fluctuations and stabilize the training process. To augment the data, all instances were subjected to random horizontal flips as well as random rotations within a certain angle range (−10° to +10°). These operations further diversify the training data and enhance the model’s ability to generalize to different scenarios.

## 4. Experimental Results

### 4.1. Evaluation Metrics

In this study, we evaluate the performance of the SOLOv2 model for the detection of high-voltage transmission lines during inspections. To assess the accuracy of the model, we use average precision (*AP*) as the evaluation metric, specifically *AP*0.50, *AP*0.75, and *AP*[0.5:0.95].

Intersection over Union (IoU) is the basis for measuring object detection accuracy. It is calculated using the following formula:(3)$$IoU=\frac{Area\;of\;Overlap}{Area\;of\;Union}$$
where the Area of Overlap represents the number of pixels where the predicted area and the ground truth area intersect. The Area of Union represents the total number of pixels covered by both the predicted area and the ground truth area.

The *AP* is computed using the following formula:(4)$$AP=\frac{TP}{TP+FP}$$
where the *TP* (True Positive) represents the number of instances correctly detected by the model, which are instances with predicted masks having an IoU exceeding the set threshold. *FP* (False Positive) refers to the number of instances incorrectly predicted by the model, which includes instances with predicted masks having an IoU lower than the threshold.

*AP*0.50 and *AP*0.75 indicate the *AP* values when the IoU threshold is set to 0.5 and 0.75, respectively. And *AP*[0.5:0.95] represents the average *AP* across different IoU thresholds ranging from 0.5 to 0.95 in steps of 0.05 (i.e., 0.5, 0.55, 0.6, 0.65, 0.7, 0.75, 0.8, 0.85, 0.9, and 0.95). This provides a comprehensive evaluation of the model’s performance over a wide range of IoU thresholds.

### 4.2. Comparison of the Components of the Proposed Framework with the Original SOLOv2

Table 1 compares the performance of the original SOLOv2 model with several improvements on the test set. The improvements mentioned include:NSCT (Non-Subsampled Contourlet Transform) preprocessing: By adding NSCT to extract and strengthen the image contour and changing the image’s original three channels into four channels, noise and redundant information in the image are effectively removed. This makes it easier for the neural network to learn these features, improving the robustness and performance of the network.Replacement of the backbone network: The original SOLOv2 network has a complex structure. In this paper, the HRNet is used as the backbone network to replace the feature extraction layer of the original network’s ResNet50 combined with FPN. HRNet can perform feature extraction and information fusion with more fine-grained features, thereby improving the model’s performance in image segmentation tasks.Addition of CBAM: CBAM is a lightweight general attention module added to the head part of the network. The CBAM attention mechanism infuses the attention map along two independent dimensions of channel and space and then multiplies the attention map with the input feature map for adaptive feature optimization, obtaining refined feature information.

Figure 5 shows the *AP* values of different models as the IoU threshold changes. As can be seen from Table 1 and Figure 5, all the proposed improvement methods can improve the model performance. However, relatively speaking, NSCT channel preprocessing improves the effect more obviously, replacing the backbone is second, and CBAM improves the model performance but is relatively small.

### 4.3. Comparison of Virtual Dataset Enhancements

In order to enhance the detection of various types of insulators, the virtual dataset is employed as a method of data enhancement. Controllable factors such as the sample category, shooting angle, environment, and presence of defects are manipulated in the virtual dataset. This approach introduces additional training data, reduces the model’s reliance on specific samples in the real dataset, mitigates the risk of overfitting, and improves the model’s ability to adapt to different input variations. Consequently, it enhances the model’s robustness and overall performance.

To assess the impact of incorporating different numbers of virtual datasets, we conducted experiments and evaluated the model’s performance. Table 2 and Figure 6 present the findings. It can be observed that utilizing the virtual dataset leads to notable improvements in the model’s performance. Within a certain range, the performance increases in line with the volume of data added. However, beyond a certain threshold, the performance gains become marginal. As shown in the table, adding a virtual dataset that roughly matches the size of the real dataset strikes a favorable balance between model performance and training time costs.

### 4.4. Comparison of the Proposed Framework with Other Approaches

The experimental results in Table 3 confirm that the improved SOLOv2 algorithm, proposed in this paper, outperforms other advanced instance segmentation algorithms such as Yolact [21], Mask RCNN [22], PolarMask [23], LSNet [24], WaveNet [25], Mask2 Former [26], CondInst [27], SOLO, and SOLOv2. Specifically, the improved SOLOv2 model achieves higher *AP*0.50, *AP*0.75, and *AP*[0.50:0.95] on the test set compared to other algorithms. When compared to the original SOLOv2 algorithm, the proposed algorithm model shows improvements of 2.3%, 1.96%, and 2.36% in *AP*50, *AP*75, and *AP*, respectively. This indicates that the improved algorithm exhibits higher accuracy and precision in detecting instances, resulting in better overall performance.

## 5. Conclusions

This paper presents an enhanced instance segmentation method using the improved SOLOv2 model for object detection in complex backgrounds, particularly focusing on high-voltage transmission component images. The proposed method improves upon the original SOLOv2 architecture by replacing the backbone network, HRNet, and feature extraction framework, FPN. This modification not only enhances the network’s ability to extract multi-scale object features but also improves the accuracy of insulator detection. To further enhance the model’s performance, we incorporate the CBAM technique, which allows for more detailed feature maps and improves the quality of masks. Additionally, we create a virtual dataset using the Unreal Engine 4 (UE4) to augment the available data. This approach not only reduces the labeling difficulty associated with real data but also increases the overall dataset size, ultimately improving the model’s detection capabilities. Experimental results demonstrate that our proposed framework outperforms other existing instance segmentation algorithms, providing a novel approach for accurate power transmission inspection in complex backgrounds.

## Figures and Tables

**Figure 1 sensors-25-05318-f001:**
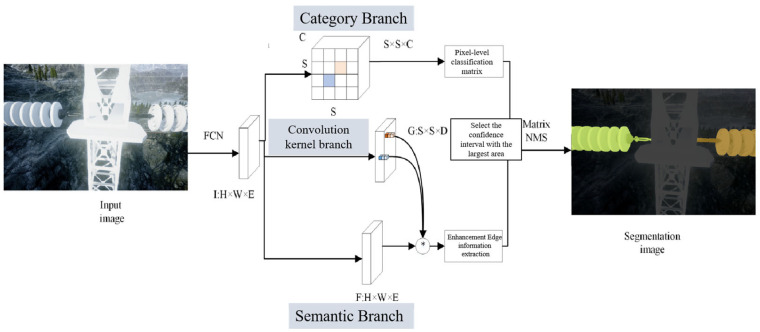
The improved SOLOv2 framework combining edge enhancement and NMS. In Semantic Branch “*” means the positional information between the two channels is multiplied to obtain the new channel information.

**Figure 2 sensors-25-05318-f002:**
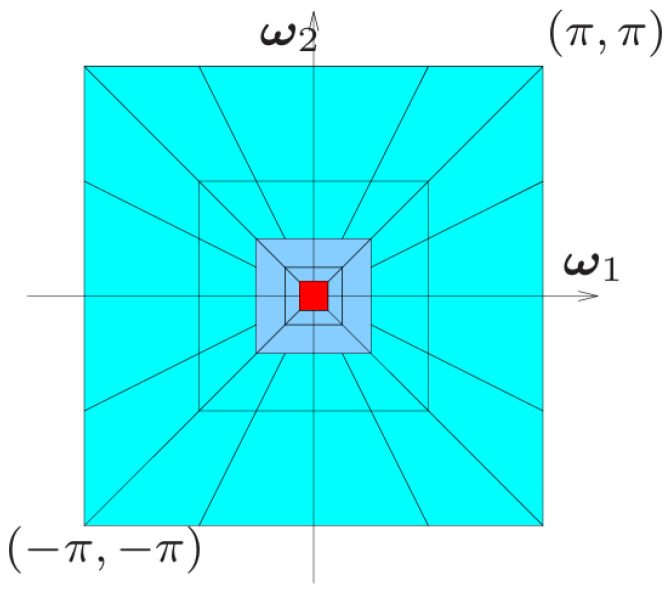
NSCT: idealized frequency partitioning obtained with the NSCT. The red area means 1/4 frequency and the Deep Blue means 1/2 frequency, Light blue means others.

**Figure 3 sensors-25-05318-f003:**
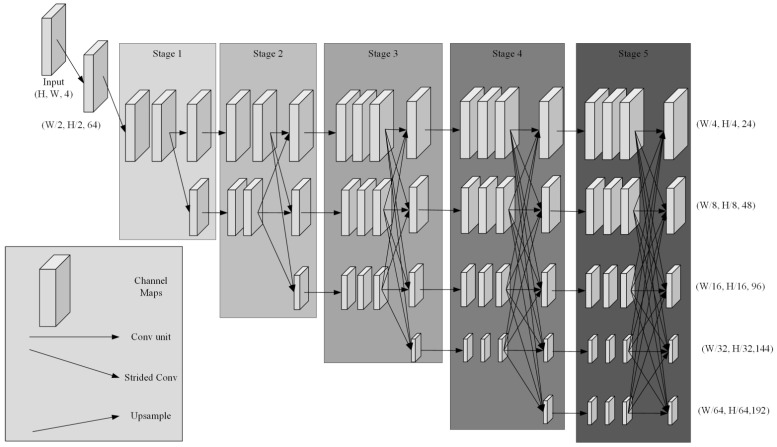
Improved HRNet structure.

**Figure 4 sensors-25-05318-f004:**
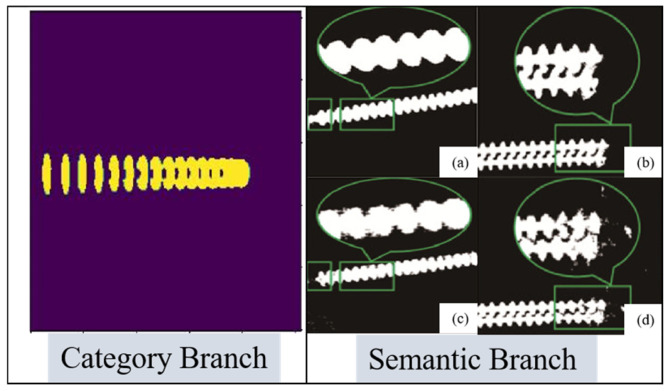
Mechanism for selecting non-maximum suppression masks. Informed by Edge-Enhanced Segmentation Data. In Semantic Branch: (**a**) WaveNet; (**b**) Mask2Former; (**c**) CondInst; (**d**) Ours model.

**Figure 5 sensors-25-05318-f005:**
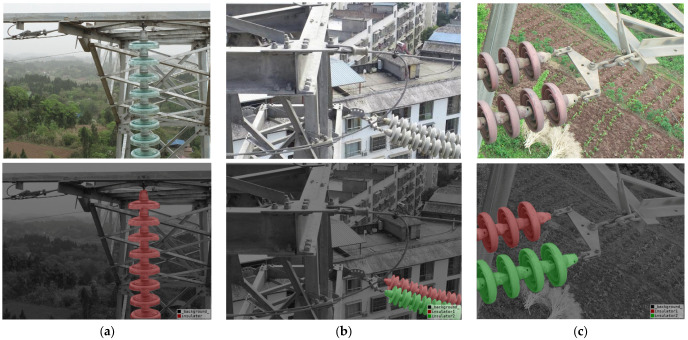
Samples of different insulators and their masks. (**a**) Wave Net; (**b**) Mask2former; (**c**) Ours.

**Figure 6 sensors-25-05318-f006:**
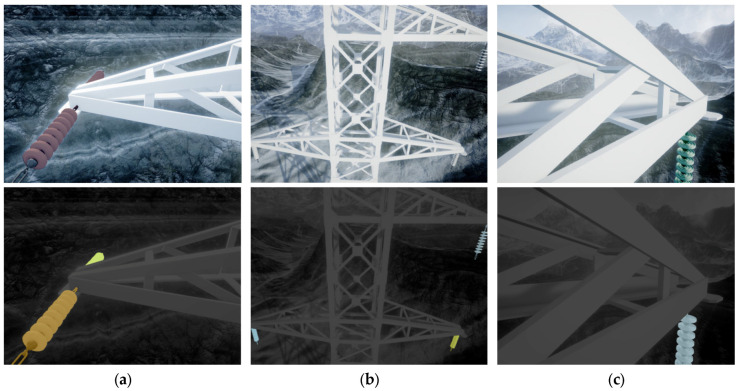
Samples of virtual dataset created by UE4. (**a**) Horizontal insulator; (**b**) Vertical insulator; (**c**) Shielding insulator.

**Figure 7 sensors-25-05318-f007:**
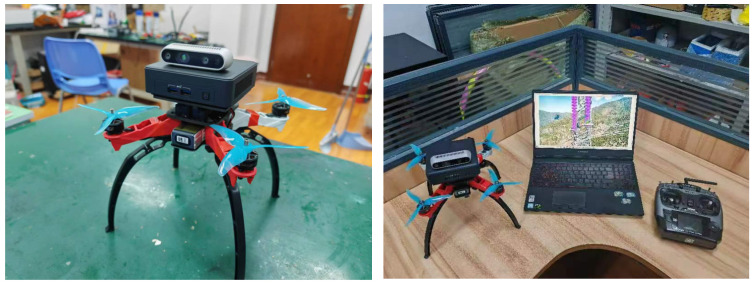
Schematic diagram of edge deployment system and cloud recognition system.

**Table 1 sensors-25-05318-t001:** Impact of different enhancement methods on model performance.

Method	*AP*0.50	*AP*0.75	*AP*[0.50:0.95]
SOLOv2	0.8791	0.8148	0.6490
SOLOv2 + HRNet	0.8848	0.8200	0.6552
SOLOv2 + CBAM	0.8832	0.8178	0.6531
SOLOv2 + NSCT	0.8817	0.8172	0.6521
SOLOv2 + HRNet + CBAM	0.8877	0.8224	0.6583
SOLOv2 + HRNet + NSCT	0.8887	0.8231	0.6592
SOLOv2 + CBAM + NSCT	0.8863	0.8201	0.6560
SOLOv2 + HRNet + CBAM + NSCT	**0.8918**	**0.8254**	**0.6623**

**Table 2 sensors-25-05318-t002:** Enhanced post-training model performance with different numbers of UE datasets.

	*AP*0.50	*AP*0.75	*AP*[0.50:0.95]
Improved SOLOv2	0.8918	0.8254	0.6623
Improved SOLOv2 + 500 virtual dataset enhance training	0.8933	0.8257	0.6635
Improved SOLOv2 + 1000 virtual dataset enhance training	0.8942	0.8270	0.6643
Improved SOLOv2 + 2000 virtual dataset enhance training	0.8959	0.8283	0.6664
Improved SOLOv2 + 3000 virtual dataset enhance training	0.8983	0.8298	0.6685
Improved SOLOv2 + 5000 virtual dataset enhance training	0.9018	0.8333	**0.6724**
Improved SOLOv2 + 7500 virtual dataset enhance training	**0.9020**	**0.8334**	0.6721
Improved SOLOv2 + 10,000 virtual dataset enhance training	0.9012	0.8332	0.6718

**Table 3 sensors-25-05318-t003:** Performances of different methods (Oous: SOLOv2 + HRNet + CBAM + NSCT).

(a) Comparison of Backbones				
Backbone	*AP*0.50	*AP*0.75	*AP*[0.50:0.95]	FPS
ResNet50	0.7908	0.7118	0.5589	20
ConvNeXt	0.8710	0.8007	0.6397	25
EfficientNet	0.8501	0.7866	0.6067	40
Vision Mamba	**0.9426**	**0.8842**	**0.7848**	8
EfficientNet-FPN	0.8791	0.8148	0.6490	30
HRNet-FPN	0.8991	0.8491	0.6691	**28**
**(b) Performances of Different Methods**				
**Model**	***AP*0.50**	***AP*0.75**	***AP*[0.50:0.95]**	**FPS**
Yolact	0.7908	0.7118	0.5589	40
Mask RCNN	0.8710	0.8007	0.6397	12
PolarMask	0.8501	0.7866	0.6067	18
LSNet	0.8426	0.7542	0.5848	42
WaveNet	0.8791	0.8148	0.6490	46
Mask2Former	0.9213	0.8541	0.6932	20
CondInst	0.8911	0.8421	0.6656	21
Ours	**0.9021**	**0.8334**	**0.6726**	**38**

## Data Availability

Due to the First Affiliation’s data privacy policy, the dataset used is not publicly available.

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
