# Peer review of "Instance Segmentation Method for Insulators in Complex Backgrounds Based on Improved SOLOv2"

_sensors, 2025, doi:10.3390/s25175318_

Round 1

Reviewer 1 Report

Comments and Suggestions for Authors

This paper proposes an insulator instance segmentation method based on an improved SOLOv2, integrating the HRNet backbone network, NSCT edge preprocessing, CBAM attention module, and UE4 virtual dataset. Although the method design is reasonable and the experimental data is complete, there are several key issues:

  1. The model design lacks innovation; what distinguishes it from existing methods? All improvements (replacement with HRNet, CBAM, NSCT, virtual data) are combinations of existing technologies, and the extent of improvement is limited (an increase of only 2.3% in AP.50).
  2. The NSCT preprocessing part lacks explanation of critical parameters (decomposition levels, number of directional filters). In the adjustment of the HRNet structure, the reduction of channels (from 384 to 144) lacks theoretical basis and experimental validation.
  3. Comparative experiments are not comprehensive; they do not compare with recent similar SOTA methods (such as Mask2Former, CondInst), only contrasting with traditional models (Mask R-CNN, YOLACT). The analysis of the domain gap between virtual and real data is insufficient.
  4. The weight λ of the loss function in equation (1) is not specified. Reference 7 is missing the journal name and volume number, and reference 8 lacks a DOI.
  5. The labeling in the structure diagram of SOLOv2 (Figure 1) is incomplete, lacking specific parameters for the dynamic convolution kernel branch. The resolution of Figure 1 is low, and the key module annotations are unclear. There is a lack of comparative diagrams showing the effect of NSCT preprocessing. The legend for the experimental result curves (such as the curve of AP varying with IOU) is not clear.
  6. The computational cost of the 1200×1600 input resolution during actual deployment has not been discussed.
  7. There is no analysis of the model's generalization ability under extreme weather conditions (such as foggy or bright light) and a lack of specialized analysis of the segmentation performance for small target insulators. Some recent research in relevant fields cited in the literature is missing. The existing relevant works on multimodal object detection can been discussed, such as LSNet, WaveNet salient, CCAFNet, ECFFNet, IRFR-Net, SSRNet KD, HKDNet crowd.
  1. Abbreviations are not expanded upon their first occurrence (e.g., NSCT). Some mathematical symbols are not in italic (e.g., IOU should be IoU). The unit formats are inconsistent (mixing "x" and "×").

Author Response

Reviewer1:

This paper proposes an insulator instance segmentation method based on an improved SOLOv2, integrating the HRNet backbone network, NSCT edge preprocessing, CBAM attention module, and UE4 virtual dataset. Although the method design is reasonable and the experimental data is complete, there are several key issues:

  1. The model design lacks innovation; what distinguishes it from existing methods? All improvements (replacement with HRNet, CBAM, NSCT, virtual data) are combinations of existing technologies, and the extent of improvement is limited (an increase of only 2.3% in AP.50).

A: Thank you very much for your guidance on our work. Based on your suggestions, we have added detailed descriptions of the improvements made to the existing SOLOv2 model, as well as the weight voting for the enhanced branches, and the composition of the focal loss. Under your guidance, the content of the work has become more complete.

  1. The NSCT preprocessing part lacks explanation of critical parameters (decomposition levels, number of directional filters). In the adjustment of the HRNet structure, the reduction of channels (from 384 to 144) lacks theoretical basis and experimental validation.

A: Your suggestions were extremely effective and helpful. We have added descriptions of stratification and filtering, as well as increased the theoretical basis by referring to similar sources. We have also rewritten the relevant sections. Thank you again.

  1. Comparative experiments are not comprehensive; they do not compare with recent similar SOTA methods (such as Mask2Former, CondInst), only contrasting with traditional models (Mask R-CNN, YOLACT). The analysis of the domain gap between virtual and real data is insufficient.

A: We have added comparisons of advanced models, included 5 new references and background writing, and re-analyzed the data. Thank you very much for your valuable feedback.

  1. The weight λ of the loss function in equation (1) is not specified. Reference 7 is missing the journal name and volume number, and reference 8 lacks a DOI.

A: Thank you very much for your detailed suggestions. We have re-explained the parameters and revised the references again.

  1. The labeling in the structure diagram of SOLOv2 (Figure 1) is incomplete, lacking specific parameters for the dynamic convolution kernel branch. The resolution of Figure 1 is low, and the key module annotations are unclear. There is a lack of comparative diagrams showing the effect of NSCT preprocessing. The legend for the experimental result curves (such as the curve of AP varying with IOU) is not clear.

A: Thank you very much for your detailed suggestions. We have revised the parameters again. At the same time, we have also re-modified Figure 1 and added descriptions for each module.

  1. The computational cost of the 1200×1600 input resolution during actual deployment has not been discussed.

A: Thank you very much for your suggestion. We have completely rewritten the entire system and deployment.

  1. There is no analysis of the model's generalization ability under extreme weather conditions (such as foggy or bright light) and a lack of specialized analysis of the segmentation performance for small target insulators. Some recent research in relevant fields cited in the literature is missing. The existing relevant works on multimodal object detection can been discussed, such as LSNet, WaveNet salient, CCAFNet, ECFFNet, IRFR-Net, SSRNet KD, HKDNet crowd.

A: Thank you very much for your suggestion. We have made comprehensive modifications to the overall numerical symbols. Based on the content of our previous work (cited), we have achieved significant effects on small targets. We have re-described them and added the work on focal loss to enhance the recognition of small targets. At the same time, we have re-compared some of the latest detection work and also added the comparison of the backbone, which has enhanced the persuasiveness.

  1. Abbreviations are not expanded upon their first occurrence (e.g., NSCT). Some mathematical symbols are not in italic (e.g., IOU should be IoU). The unit formats are inconsistent (mixing "x" and "×").

A: Thank you very much for your detailed suggestions. We have re-explained the parameters and revised the references again.

Reviewer 2 Report

Comments and Suggestions for Authors

The paper presents a solution for segmentation of high-voltage insulators (which are enough regular shape) in complex background. The authors rely on SOLO-v2 architecture and add one more branch referred to as 'category'. Experiments are performed with real and synthetic data against several alternative solutions. The paper has the following drawbacks. 

1. As said, the third branch is added, outputting 'semantic category' data. Obtaining these data is described and is understandable. Two original branches of SOLO-v2 yield 'Instance mask', that is also understandable. The these two type of data are merged finally to form 'matrix NMS'. However the process of merger is not described. Please add complete description. 

2. Only ResNet50-based solutions are considered as alternatives. Why? There are plenty of other solutions for image semantic segmentation. If they are not adequate, please explain. If they can be applied please test at least some of them. 

Minors: 

1. Check parentheses balance in line 113.

2. line 114: "is 1/4" -> "are 1/4". 

3. Figure 1. Signs 'Feature branch' and 'Category branch' are not placed well. What operation is done to obtain "Matrix NMS' from two inputs? 

Author Response

Reviewer2:

The paper presents a solution for segmentation of high-voltage insulators (which are enough regular shape) in complex background. The authors rely on SOLO-v2 architecture and add one more branch referred to as 'category'. Experiments are performed with real and synthetic data against several alternative solutions. The paper has the following drawbacks. 

  1. As said, the third branch is added, outputting 'semantic category' data. Obtaining these data is described and is understandable. Two original branches of SOLO-v2 yield 'Instance mask', that is also understandable. The these two type of data are merged finally to form 'matrix NMS'. However the process of merger is not described. Please add complete description. 

A: We have completely rewritten this section and have also added similar descriptions of NMS.

  1. Only ResNet50-based solutions are considered as alternatives. Why? There are plenty of other solutions for image semantic segmentation. If they are not adequate, please explain. If they can be applied please test at least some of them. 

A: We have included the latest version of Backbone and the most recent research findings for comparison, as well as added descriptions of experiments and systems. Thank you very much.

Minors: 

  1. Check parentheses balance in line 113.

A: Thank you very much for your feedback. We have made revisions to the entire text.

  1. line 114: "is 1/4" -> "are 1/4". 

A: Thank you very much for your feedback. We have made revisions to the entire text.

  1. Figure 1. Signs 'Feature branch' and 'Category branch' are not placed well. What operation is done to obtain "Matrix NMS' from two inputs? 

A: Thank you very much for your feedback. We have made revisions to the entire text. At the same time, the process of NMS and the description of the branches have been added.

Round 2

Reviewer 2 Report

Comments and Suggestions for Authors

The authors have responded to comments and revised the paper accordingly making it quality enough for publication. Except one case: Figure 1 contains three branches, but now there are four signs to these branches. Please remove the excessive one. 

Author Response

Thank you very much for your suggestion. We have made the necessary changes to the description in the picture according to your advice. We sincerely hope that you will continue to offer us your help and suggestions.
